# Fronto–Cerebellar Diaschisis and Cognitive Dysfunction after Pontine Stroke: A Case Series and Systematic Review

**DOI:** 10.3390/biomedicines12030623

**Published:** 2024-03-11

**Authors:** Kei Shimmyo, Shigeru Obayashi

**Affiliations:** Department of Rehabilitation Medicine, Saitama Medical Center, Saitama Medical University, 1981 Kamoda, Kawagoe 350-8550, Japan; kshimmyo@saitama-med.ac.jp

**Keywords:** brainstem, cerebellum, cognition, diaschisis, executive function, functional NIRS, near-infrared spectroscopy, SPECT, supplementary motor area, thalamus

## Abstract

It is well known that cortical damage may affect cognitive functions, whereas subcortical damage, especially brainstem stroke, would be far less likely to cause cognitive decline, resulting in this condition being overlooked. Few studies have focused on cognitive dysfunction after a pontine stroke. Here, we begin with describing our nine new case reports of in-depth neuropsychological findings from patients with pontine stroke. The dominant domain of cognitive dysfunction was commonly characterized by executive dysfunction, almost in line with previous studies. The severity was relatively mild. We give an overview of the available literature on cognitive decline following a pontine stroke. This is followed by discussions regarding the prognosis of the cognitive disabilities. Based on previous neuroimaging findings, we would like to get to the core of the neuropathology underlying the cognitive declines in the context of “diaschisis”, a phenomenon of a broad range of brain dysfunctions remote from the local lesions. Specifically, our unique paper, with two modalities of neuroimaging techniques, may help us better understand the pathology. SPECT scans yield evidence of frontal and thalamic hyper-perfusion and cerebellar hypo-perfusion in patients with pontine stroke. Functional near-infrared spectroscopy, when focusing on the supplementary motor area (SMA) as one of the hyper-perfusion areas, exhibits that SMA responses may be subject to the severity of cognitive decline due to a pontine stroke and would also be related to the recovery. Finally, we posit that cognitive decline due to pontine stroke could be explained by the failure of hierarchical cognitive processing in the fronto–ponto–cerebellar–thalamic loop.

## 1. Introduction

Stroke not only causes functional disabilities, such as motor paresis and dependency in activities of daily living (ADL), but also affects cognition as an invisible disability [1,2,3,4,5,6]. Due to this invisibility, the cognitive impairment inherent in stroke survivors has often been overlooked during follow-ups. As clinical determinants, the prevalence of cognitive impairment differs according to the lesion location, such as cortical, subcortical, and infratentorial lesions (cerebellum and brainstem) [7,8,9,10,11,12,13]. A previous study demonstrated that cognitive impairment was present in 74% of acute-phase patients with a cortical stroke, 46% with a subcortical one, and 43% with an infratentorial one [14]. Insights into the modulatory role of the cerebellum in cognition are well documented [15,16,17,18]. On the other hand, conventional tenets posit that damage to the brainstem might not affect cognition. To date, the contribution of the brainstem in cognition is still underexplored [19,20,21,22,23,24,25,26,27]. Also, the prevalence, severity, and long-term trajectories of cognitive dysfunctions due to a brainstem stroke remain unknown. The literature is somewhat limited and lacks considerable data because of relatively small sample sizes in the relevant studies [28,29,30,31,32]. To the best of our knowledge, we only found a review article of this topic [19]. In addition, most of the cases in the literature have not always achieved sufficient neuropsychological outcome measures to document the broad range of cognitive impairments possible after a brainstem stroke. Pontine infarction accounts for about seven percent of all ischemic strokes [33,34]. Since most cases have not been limited to an isolated pontine stroke, we could not distinguish whether the type and severity of the cognitive symptom depends on the location of the injury. Accordingly, we focused on the pons to explore the cognitive function. An in-depth neuropsychological evaluation of the case series of pontine strokes could disclose the characteristics of the cognitive disabilities, which will help physicians and healthcare professionals better understand the deficits.

How do cognitive dysfunctions arise from subcortical or infratentorial lesions? The possible mechanism may be explained by the theory of “diaschisis”, where the impact of focal lesions can expand into a widespread and diffuse brain network organization, remote from the lesion location [35,36,37,38]. Neuroimaging techniques would be suitable tools for visualizing the mechanisms underlying invisible disabilities in the form of diaschisis. In favor of this view, SPECT scans detected cerebral perfusion abnormalities or decreases in the regional cerebral blood flow in remote brain regions after a brainstem stroke [20,22,23,39,40]. Intriguingly, the injury is followed by morphological and degenerative changes in the brain, such as frontal and thalamic volume expansions or cerebellar atrophy [41,42,43,44], anterograde and retrograde degeneration in the corticospinal tracts [45,46,47], and aberrant functional connectivity [48,49,50]. 

The merit of the present case report lies in the fact that it helps the physicians become aware of cognitive decline after a pontine stroke and that further understanding of the neuropathology may eventually lead to the conquest of cognitive dysfunction. Now, we begin with the case series description by sharing the in-depth neuropsychological findings for nine patients with pontine stroke and provide clear definitions of the neuropsychological profiles. And we give an overview of the available literature on cognitive decline following a pontine stroke. The severity and prognosis of the cognitive decline is also discussed. Then, we will get to the main subject: all-encompassing deliberations about the neuropathology of the cognitive deficits based on neuroimaging findings and the theory of “diaschisis”. While reviewing the literature dealing with the possible pathology of cognitive dysfunction, especially involving the frontal lobe and the cerebellum, we focus on the significant role of the fronto–ponto–cerebellar–thalamic loop in the neuropathology based on our recent data from two modalities of neuroimaging techniques, SPECT, and functional near-infrared spectroscopy (f-NIRS) [23]. Finally, we propose a specific function for pons in the hierarchical information processing system of this loop. 

## 2. Case Series Description

The inclusion criteria were as follows: (1) under the age of 90; (2) first-ever isolated pontine infarct; (3) adequate mental state to participate (clear consciousness); (4) medically stable condition; (5) within two weeks of stroke onset at first time of the evaluation; and (6) independent ADL before admission. The exclusion criteria were as follows: (1) history of damage from a stroke (cerebral infarct, cerebral hemorrhage, subarachnoid hemorrhage, and lacunar infarct), brain injury, or brain tumor; (2) neurodegenerative disease; (3) mental illness; (4) dementia; (5) epilepsy; and (6) severe or moderate hemiparesis. Among a total of 163 cases, 9 cases of pontine stroke were selected, according to the inclusion and exclusion criteria. These cases consisted of seven men and two women with a mean age of 76.44 years and a range between 63 and 86 years (Table 1). In terms of the stroke type, the case series consisted of seven patients with pontine branch atheromatous disease (BAD), one patient with a paramedian lacunar infarct, and a one with a pontine hemorrhage. The stroke location was determined using an MRI scan or CT scan of the brain (Figure 1). Apart from the stroke in the pons, no additional lesions or atrophy were detected. Additionally, our case series was narratively described. All patients were assessed by well-trained neuropsychologists using the standardized Japanese translation of the Mini-Mental State Examination (MMSE) for the general intellectual ability [51] and a set of neuropsychological test batteries for attention, memory, and executive function. The batteries consisted of the following tests: the Trail Making Test [52,53], Japanese version-A (TMT-J part A), for assessing attention and processing speed; the TMT-B (Japanese version, where Kana letters replaced the Roman alphabet) and the ΔTMT, the temporal gap between B and A and the Frontal Assessment Battery (FAB) [54], for assessing the executive function; and the Standard Verbal Paired-Associate Learning Test (S-PA) [55] and Rey–Osterrieth Complex Figure Test (ROCFT) [56], for assessing the verbal memory and visual memory, respectively. Also, the reproduction of the ROCFT requires some strategy in term of its accuracy, involving executive function and visuospatial cognition. Abnormalities of both the TMT-J and S-PA were determined based on a database of normal healthy volunteers (mean and SD for people in their 60s and 70s). The Brunnstrom recovery stage (BRS) is designed to describe the motor recovery process of a sequence of limbs as well as the severity of hemiparesis, containing three items for the arm (shoulder/elbow/forearm: BRS-A), the hand/finger (BRS-H), and the leg (BRS-L), all of which are rated on a six-level scale (level 1 to 6) [57]. Subjects provided written informed consent after receiving a detailed explanation of the procedures. This study was reviewed and approved by the Ethics Committee of Saitama Medical Center (Approval number: 2021-093).

### 2.1. Case 1

A 63-year-old male, with a history of hypertension, developed right mild hemiparesis (BRS-A 6, BRS-H 5, and BRS-L 6) and dysarthria. Three days after onset, he walked into our hospital with his feet dragging. MRI imaging of his brain revealed a left ventral pontine infarction, which was considered to be BAD (branch atheromatous disease). He received conservative treatment with dual antiplatelet therapy (DAPT). Three days after admission, a neuropsychological investigation was started. The MMSE produced a normal score of 29/30, guaranteeing his preserved orientation. His executive function was mildly impaired, as he made a mistake on the similarities and lexical fluency, while he scored 16 on the FAB. The TMT-J -A response (43 s) was within the normal range, whereas the TMT-B (106 s) produced delayed responses, and the ΔTMT was gapped by 63 s, revealing his dysexecutive function syndrome. His verbal memory disturbance was delineated by the S-PA. A follow-up of the TMT, conducted five days after the initial test, showed an improvement in the ΔTMT to a normal range. Seven days after admission, he regained his hemiparesis fully and was discharged to his home.

### 2.2. Case 2

A 72-year-old male developed right mild hemiparesis (BRS-A 6, BRS-H 6, and BRS-L 6), numbness, and facial paralysis. The risk factors for stroke were hypertension and dyslipidemia. On the day of onset, he was admitted to the hospital, and MRI imaging of the brain revealed left pontine BAD. He received conservative treatment with DAPT, an anticoagulant and an antioxidant drug. On the day after admission, he complained of diplopia. At four days after admission, a neuropsychological investigation was started. He scored 26/30 on the MMSE, with points lost in Serial sevens test. Executive dysfunction was evident from the fact that he made a mistake in the Go/No-Go test on the FAB, and the reproduction of the ROCFT scored 31/36. The TMT-J was delayed in both A and B at 110 and 156 s, respectively. The S-PA was estimated to be within the normal range. Nine days after admission, he was transferred to the rehabilitation hospital.

### 2.3. Case 3 

A 75-year-old female developed right hemiparesis (BRS-A 3, BRS-H 3, and BRS-L 4), diplopia, and dysarthria. One day after onset, she was admitted to our hospital. MRI imaging of the brain revealed isolated pontine BAD. She received conservative treatment with DAPT, an anticoagulant and an antioxidant drug. Two days after admission, she scored 29 on the MMSE. Her TMT-A response was at a normal level, but she had delayed TMT-B responses, showing a significant gap (ΔTMT 74 s), and low scores for the similarities and lexical fluency in the FAB revealed her dysexecutive function syndrome. A mild memory dysfunction was proven by the S-PA, with a delayed recall on the MMSE and ROCFT. At 17 days after the first assessment, a follow-up of the TMT-J recovered to normal range (A-39 s/B-69 s). At 27 days after admission, she transferred to the rehabilitation hospital, because her hemiparesis (BRS-A 3, BRS-H 3, and BRS-L 4) and dysarthria persisted.

### 2.4. Case 4

A 76-year-old male lost consciousness during work and was taken to the emergency room of our hospital. He regained consciousness and showed dysarthria and right medial longitudinal fasciculus syndrome. He had a history of myocardial infarction, and he carried an implanted cardioverter defibrillator in his body. A CT scan showed a high-density area at the top of the basilar artery. After he underwent tissue plasminogen activator (t-PA) therapy and an endovascular therapy with mechanical thrombectomy, his eye movement symptom disappeared. One day after admission, the dysarthria persisted, and the CT scan displayed a tiny low-density area isolated at the paramedian pons. He received conservative treatment with a direct oral anticoagulant (DOAC). Two days after admission, he made a mistake on the recall and three-stage command in the MMSE (resulting in a total score of 27). Lower S-PA scores and a delayed recall in the ROCFT revealed his mild verbal and visual memory deficits, respectively. He had a response delay in the TMT-A (118 s), showing an attention deficit. His executive dysfunction was evident using the TMT-B (292 s), showing a gap (ΔTMT 174 s), and he had low scores for lexical fluency and Go/No-Go on the FAB. At 12 days after the first assessment, a follow-up of the MMSE and FAB revealed no score changes relative to the first assessment. He was discharged to his home 16 days after admission.

### 2.5. Case 5 

A 77-year-old male developed left hemiparesis (BRS-A 4, BRS-H 4, and BRS-L 5), facial palsy, and dysarthria. The risk factors for stroke were hypertension and sleep apnea syndrome. An MRI showed right pontine BAD. He received conservative treatment with DAPT, an anticoagulant and an antioxidant drug. His MMSE score was 28, guaranteeing his preserved orientation. Three days after admission, although within the normal limit of the TMT-A (60 s), he exhibited a significant gap between TMT-A and B (ΔTMT 149 s) and had low scores for the similarities and lexical fluency on the FAB, suggesting executive dysfunction. Copy of the ROCFT presented with his low scores, suggesting executive dysfunction and visuospatial recognition difficulty, and the S-PA revealed a normal verbal memory. He recovered from the left hemiparesis fully and was discharged to his home 30 days after admission.

### 2.6. Case 6 

A 77-year-old female developed left facial numbness, dysarthria, and right abducens nerve palsy. The risk factor was diabetes mellitus. A CT scan showed a high-density area isolated at the right dorsal pons, diagnosing as pontine hemorrhage. The day after admission, a mild right hemiparesis (BRS-A 6, BRS-H 6, and BRS-L 6) was found. She received conservative treatment with an antihypertensive drug. Her MMSE score was 28, guaranteeing her good orientation. Three days after admission, low scores for similarities, lexical fluency, and the Go/No-Go test on the FAB suggested executive dysfunction. She showed normal responses to the TMT (A-60 s/B-79 s). A low score (11/36) of delayed recall in the ROCFT showed her visual memory disorder, while the S-PA displayed normal scores. Her numbness persisted, but she recovered from hemiparesis, dysarthria, and abducens nerve palsy. She was discharged 17 days after admission.

### 2.7. Case 7

A 79-year-old male developed dysarthria. His past history revealed microscopic polyangiitis. An MRI of the brain indicated left pontine BAD. He received conservative treatment with DAPT, an anticoagulant and an antioxidant drug. His MMSE score was 25, which included mistakes in attention and calculation, delayed recall, and picture copy. Three days after onset, he exhibited difficulties in similarities, lexical fluency, and the Go/No-Go task on the FAB, showing executive dysfunction. He had delayed TMT-A responses (84 s), indicating his attention deficits. A low score (16/36) of delayed recall in the ROCFT proved his visual memory impairment. He recovered from dysarthria and was discharged 42 days after admission.

### 2.8. Case 8

An 82-year-old male developed left mild hemiparesis (BRS-A 5, BRS-H 5, and BRS-L 5) on the day of onset, and MRI imaging of the brain revealed right pontine BAD. After admission, dysphagia was also revealed. He received conservative treatment with DAPT, an anticoagulant and an antioxidant drug. At two days after admission, his MMSE score was 23/30 (a total of 7 points were lost in Serial sevens and delayed recall), although guaranteeing his preserved orientation. It suggested his inattention and memory disturbance. He made a mistake on the similarities, lexical fluency, and Go/No-Go test, and his FAB score summed up to 10/18, suggesting his executive dysfunction. The response for TMT-A was delayed (243 s), suggesting his attention deficit. Because of his persistent hemiparesis, he was transferred to the rehabilitation hospital at eight days after admission.

### 2.9. Case 9

An 86-year-old female with independent activities of living developed right hemiparesis (BRS-A 2, BRS-H 2, and BRS-L 4), left facial palsy, and dysarthria. A risk factor for stroke was hypertension, and her past history included cardiac angina and skin cancer. An MRI revealed left pontine BAD. She received conservative treatment with DAPT, an anticoagulant and an antioxidant drug. Two days after admission, her MMSE had a full score of 30. Her dysexecutive function syndrome was evident in the low scores of similarities and lexical fluency on the FAB. Examinations of other domains were not available. Because of her persistent hemiparesis (BRS-A 3, BRS-H 4, and BRS-L 3), she was transferred to the rehabilitation hospital one month after admission.

## 3. Summary of Clinical Data and Review of the Literature

### 3.1. Summary of Case Series

Six patients presented with mild hemiparesis and eight patients showed dysarthria. Nine new cases of cognitive declines after a pontine stroke were documented. The domain-specific frequencies of cognitive dysfunction after a pontine stroke were delineated as follows: executive dysfunction (9/9 cases), visual memory disturbance (6/8), verbal memory disorder (2/8), and attention deficits (4/8) (Table 2). The dominant domains were characterized by a dysexecutive function and visual memory impairment, almost in line with previous studies. The severity was relatively mild. Short-term follow-up investigations were achieved in three cases, two in which the patients recovered from executive dysfunction following pons injury and the other with persistent cognitive decline. A long-term follow-up was not available for any patients. An overview from previous case studies, commonly demonstrating executive dysfunction as the most frequent domain, may support the above view. However, this domain tendency may be slightly different from our previous study [23], mainly because the latter was relatively lacking in in-depth neuropsychological investigations. An initial screen identified abstracts or titles. The second screening was based on the full-text review. Two investigators (KS and SO) independently assessed the full text for eligibility; discrepancies were resolved via discussion.

### 3.2. Literature Search and Study Eligibility

Online article databases including PubMed, MEDLINE, and Scopus were searched to identify the relevant literature on cases describing cognitive disturbances following brainstem stroke lesions. Cases with brain traumas, infections, or primary neurodegenerative diseases were excluded in this study, because they are typically associated with more widespread brain damage than an isolated brain injury, and they also often present functional deficits not limited to the lesion itself. The following keywords were used to search the electronic databases: *cognitive dysfunction, cognitive impairment, brainstem, pons, and pontine.* Only articles that provided details regarding the cases describing neurological tests were included in the overview. The articles were dated from 1998 until March 2023.

The initial literature search of keywords produced 345 results. Due to the inclusion criteria, 264 studies were excluded. Among the 19 studies screened for eligibility, 9 met our criteria. A flow chart summarizing the selection process is depicted in Figure 2. 

### 3.3. Methodological Quality and Risk of Bias

When undertaking a systematic review while considering the strengths and weaknesses of the research, the risk of bias in nonrandomized studies of interventions must be assessed. The Risk of Bias Assessment Tool for Nonrandomized Studies (RoBANS) version 2 is a comprehensive checklist instrument for assessing the risk of bias in cohort studies, case-control studies, and cross-sectional studies [58]. Case-control studies in the included papers were applied (Table 3). On the contrary, case reports and case series are uncontrolled study designs known for their increased risk of bias, but they have profoundly influenced the literature and continue to advance our knowledge. Murad presented a framework to evaluate the methodological qualities of case reports and case series based on the domains of selection, ascertainment, causality, and reporting [59]. The methodological quality of the included case reports was evaluated using this method (Table 4).

### 3.4. Synthesis

Owing to the study heterogeneity in terms of the quantitative designs and methods, a quantitative synthesis was not feasible in this systematic review. A narrative approach to synthesis was applied, including textual descriptions, tabulation, grouping, and content analysis for data translation [65]. One author (KS) carried out the initial synthesis, and iterative discussions with the other author (SO) were conducted to refine the essential elements of this review.

### 3.5. Characteristics of Cognitive Impairment after Pontine Stroke

There are some evidential items from group analyses that were determined by comparing the pontine stroke patients and healthy controls. Van Zandvoort et al. described the cognitive dysfunction of 17 patients with a brainstem stroke (including 13 with pons lesions) [60]. The brainstem group had significantly impaired language (naming), category fluencies, attention, executive functions, and visuospatial abilities compared with the age-matched control group. Wang et al. compared a set of neuropsychological results between 47 pontine stroke patients and 55 age-matched healthy control subjects. The pons lesion group had a significantly weakened executive function, working memory, and spatial memory relative to the healthy control group [48]. Fu et al. compared neuropsychological data between 34 brainstem stroke patients and a healthy control group. In regard to the distribution of the lesions, the mesencephalon was most common, followed by the pons. Significant differences in attention, visuospatial abilities, and language can be discerned in the brainstem lesion group [61]. 

So far, several individual details of the disabilities have been so far reported. D’aes and Mariën analyzed the cognitive characteristics of a brainstem stroke in 33 patients. This included 22 pontine stroke victims who exhibited executive dysfunctions, memory disturbances, and attention deficits [19]. They claimed that the most frequent domain was that of executive dysfunction, followed by inattention. For further details, Hoffmann and Watts reported that all four pontine stroke patients had executive dysfunction, such as disabilities in planning, initiating, and executing activities, as well as impaired self-monitoring [22]. Another case report demonstrated executive dysfunction [26]. Gerrard et al. reported that a pontine stroke generated executive dysfunction (3/4 patients), attention deficits (3/4), and memory disturbances (1/4) [21]. Four months of follow-up delineated persistent attention deficits and executive dysfunctions. All domain-specific outcomes were likely to be less severe and more transient than supratentorial cortical infarcts, which was also consistent with a previous study [37]. Obayashi described neuropsychological findings from 25 patients [23]. They presented attention deficits (20/25 patients), memory disturbances (15/25), executive dysfunctions (15/25), and social behavioral disturbances (1/25), with one cognitive domain for four patients, six for two domains, and seven for three domains [23]. A few months of follow-ups was achieved for two patients, who showed improvements in executive dysfunction. Nishio et al. [62] described a single case of a 74-year-old female with a pontine infarct who developed attention deficits and dysexecutive function syndrome while her memory, language, and visuospatial abilities remained intact. Maeshima et al. [63] reported a case of a 54 year-old male with a pontine hemorrhage presenting with executive dysfunction, attention deficits, and memory disturbance. Six months after onset, all domains were recovered. Neki et al. [64] demonstrated domain-specific impairments after a pontine hemorrhage. Five out of ten patients met the inclusion criteria, all of which affected the executive function. However, other domains were not investigated, except for the general intelligence ability.

The overall results of the included studies are summarized in Table 5. 

## 4. Discussion

Conventionally, physicians and healthcare professionals posit that damage to the brainstem might not affect cognition. The aim of this paper is to make clear whether a pontine stroke could affect the cognition of patients and to disclose the domain characteristics while providing an overview of our case series and previous studies relevant to cognitive decline after a pontine stroke. Nine new cases of cognitive declines poststroke were commonly characterized by executive dysfunction, almost in line with previous studies. The severity was relatively mild. This trait is supported by a previous study exhibiting prefrontal dysfunction after brainstem damage [36] and is also based on anatomical evidence of the frontopontine pathways from the frontal association areas [66]. The ultimate goal of this paper is to disclose the presence and characteristics of cognitive impairments due to pontine stroke and to help the patients reintegrate into society. Awareness of the disabilities will enlighten physicians in clinical practice and likely improve clinical diagnostics and patient care. The notion that the damage to the pontine affects cognition may have a significant impact on rehabilitation interventions, usually assuming that all attention in the intervention should be directed toward pure sensorimotor recovery for patients with an isolated pontine stroke. It will be better if physicians and healthcare professionals decide more carefully whether the patient can return to work or drive a car. In the coming years, this research focus will be critical for further understanding the specific functions of the pons in cognition and the neuropathology of underlying cognitive deficits due to pontine injury. It will provide new insights into the neurobiology of cognition and develop new treatment and management strategies for cognitive impairment after a stroke. To achieve this goal, the use of non-invasive neuroimaging techniques would be very powerful and beneficial. Now, we highlight neuropsychological findings from our nine new case studies as well as the relevant studies, and as mentioned below, the literature review will get to the core of how the application of multi-modal neuroimaging techniques will expand our understanding of the neural mechanisms responsible for cognitive disabilities following a pontine stroke.

### 4.1. Neurobiology of Executive Function, Attention, and Memory

It is no wonder that the pathophysiology of cognitive impairment results from the disruptions of neural mechanisms underlying cognitive domains, such as executive function, attention, and memory. Basically, executive function is composed of self-control, self-monitoring, emotional control, flexibility, task initiation, organization, working memory, and planning and time management, so that the achievement of these functions may involve a widely distributed cortical network [67,68,69]. The function is responsible for the inferior frontal cortex (IFG), basal ganglia, and pre-supplementary motor area (pre-SMA) [70] or anterior cingulate cortex and parietal cortex [71], as well as the cerebellum [15]. Accordingly, these brain regions may be related to dysexecutive function after pontine injury. The molecular mechanism of executive function remains unknown, but an intriguing study has been reported. The paper stated the relevance of C-C chemokine receptor 5 (CCR5) expression in the cortex for cognitive recovery and motor recovery [72]. The authors claimed that CCR5 is upregulated in the cerebral cortex after a stroke and traumatic brain injury, and that spatial working memory disturbances due to traumatic brain injury were improved by the inhibition of CCR5, suggesting that cognitive recovery may be modulated by CCR5 expression.

On the other hand, models of attention have been postulated [73,74,75,76], but the definition of attention itself remains unclear. Posner proposed a hypothesis that attention may be composed of the alerting system, the orienting system, and the executive control of attention [77,78,79]. The neural mechanism for attention can be separated by three systems: (1) alerting that produces and maintains optimal vigilance, (2) orienting that prioritizes sensory input by selecting a modality or location, and (3) executive control that involves task switching, initiation, adjustments, and maintenance within trials in real time. The alerting system is attributed to the right frontal cortex and right parietal cortex. The orienting system involves the superior parietal and frontal lobes, and the executive control of attention is subserved by the frontoparietal system and cingulo–opercular system [80]. So, attention and executive function cannot be separated from each other. This suggests that attentional processing might be relatively fundamental and is involved in a broader range of brain areas than executive function. Moreover, it is possible that pontine damage is more likely to generate executive dysfunction and attentional deficits. 

The mechanism of memory has been extensively studied in greater detail than other domains. A great deal about the cellular and molecular mechanisms of long-term memory storage has been learned at the level of the synapse [81,82,83,84], but the mechanism of consolidation at the level of neuronal systems has been relatively overlooked [85,86]. Episodic memory refers to a declarative memory that contains information specific to the time and place of acquisition [87]. Episodic memory retrieval is attributable to the frontal cortex, posterior parietal cortex, and medial temporal cortex [88], whereas semantic memory retrieval is responsible for the top–down signal from the prefrontal cortex and the subsequent memory processing of perirhinal circuits as a storage of memory in a hierarchical manner [89]. In contrast, procedural memory is achieved by the activation transition of the fronto–parieto–cerebellar circuit, such that the dorsolateral PFC and pre-SMA engage at an early stage of learning, the parietal IPS and precuneus work at an intermediate stage, and the cerebellum serves as storage at the final stage [90,91,92,93]. Also, the fronto–parieto–cerebellar circuit through the corpus callosum may contribute to the inter-manual transfer of procedural memory [92,93].

### 4.2. Short- and Long-Term Changes in Cognitive Decline Poststroke

The poststroke cognitive function changes temporally and dynamically over time. However, details of the longitudinal trajectory of domain-specific cognitive alterations after a stroke remain unknown [94,95,96]. Until now, few studies have dealt with the longitudinal effect of stroke on cognition, and they showed mixed results, either a trend toward deterioration [97], persistence [98,99], or improvements [100,101]. A recent study reported the probability of poststroke cognitive declines and follow-up alterations [101]. Cognitive impairment was present in 59% of survivors at three months poststroke, and 51% remained at eighteen months after onset. Some domains, such as executive function and language, improved during the follow-up period, but it is difficult to determine which of the cognitive domains was more inclined to recovery. Other reports suggested that executive and language functions as well as the visuospatial function may improve 3 months to 1.5 years after a stroke [101,102,103], while another report described that working memory may eventually recover years after a stroke [23]. In addition, the speed of recovery may differ depending on the lesion location. In the case of younger stroke survivors, domain-specific cognitive impairments improved or were stable 10 years after their stroke [102]. On the other hand, stroke is associated with an increased risk of dementia [104,105,106,107]. A previous report demonstrated about a twofold increase in cognitive decline after a stroke relative to before the stroke [94]. The executive performance has also been reported to be an excellent predictor of vascular dementia [108,109]. Most stroke survivors may fully recover from the decline between 3 and 15 months afterward [110,111], but others do not recover [112] and deteriorate to dementia. Likewise, a systematic review focusing on the natural history of cognitive impairment after a stroke replicated the mixed results [103]. As mentioned above, however, our observation and the relevant literature may seemingly imply the relatively early regaining of cognitive functions after pontine stroke. This view would be supported by the pathophysiology of cognitive decline after a pontine stroke in terms of “diaschisis”.

### 4.3. Profile of “Diaschisis”

“Diaschisis” is well known as a phenomenon consisting of a broad range of depressed brain functions, which are remote from local lesions of the central nervous system. Age negatively influences the severity of diaschisis and determines the extent to which the patient recovers. The older the patient after a stroke, the more severe the neurological deficit and the less complete the neurological recovery. In addition, the corpus callosum plays a crucial role in the remote effects of diaschisis [113]. A previous study suggested that the deficits arising from infratentorial infarcts tended to be less severe and more transient than those from supratentorial cortical infarcts [37]. At the very least, given that cognitive decline after pontine injury represents cerebro-cerebellar diaschisis, it is expected that the decline would be less disabling and would eventually recover.

### 4.4. Neuroimaging of “Diaschisis”: Single Photon Emission Tomography (SPECT) 

A previous study reported that 58% of 55 patients with a supratentorial stroke presented with cerebellar hypoperfusion [114]. Some SPECT studies yielded evidence of cerebellar hypoperfusion after a pontine stroke [115] and lateral medullary infarcts [116]. Conversely, patients with a unilateral cerebellar stroke revealed contralateral cerebral hypoperfusion [117]. A previous SPECT study demonstrated that patients with a brainstem stroke presented an aberrant perfusion pattern in the ipsilateral frontoparietal lobes and the contralateral cerebellum, as evidence of diaschisis [20]. Another SPECT study reported frontal and parietal hypoperfusion in patients with brainstem infarcts [22]. The possible mechanism of the remote effects may be explained by the reciprocal connections of cortico-pontine-cerebellar fibers, which in turn project to the red nucleus and ventrolateral nucleus of the thalamus and to the frontoparietal cortex. Our recent SPECT study delineated that the patients in the acute phase of a pontine infarct showed hyper-perfusion in the bilateral frontal cortices, parietal cortices, and right thalamus and hypo-perfusion in bilateral cerebellum [23]. For more details regarding our earlier report [23], in our previously reported Case 3 (not the present case 3 mentioned above), SPECT revealed decreases in the right Brodmann area (BA) 39 and right putamen, and increases in the bilateral BAs 6 and 8, 44, bilateral 40, bilateral BA 24s and 32, and left putamen. In an earlier Case 4 from the report, SPECT revealed decreases in the left BAs 44 and 45, bilateral BAs 24 and 32, and bilateral putamen, and increases in the right BAs 6 and 8, BA 45, right BA 24, and bilateral BAs 39 and 40. In the earlier Case 5, SPECT showed decreases in the bilateral putamen, left BA 32, and right BA 39, and increases in the bilateral BAs 6 and 8, BAs 44 and 45, right BA 24, and bilateral BA 40. In the earlier Case 7, SPECT revealed decreases in the left BA 39, 40, and left putamen, and increases in the bilateral BA 6, 8, 44, right BA 39, 40, right BA 24, and bilateral putamen. It is very likely that depressed brain areas mainly represent vascular alterations of the diaschisis phenomenon in terms of neuro-vascular coupling, while hyper-perfusion brain areas largely reflect the compensating process for cognitive decline after a pontine stroke. Possible mechanisms for inter-subjective differences in perfusion abnormalities may be explained by the following: (1) differences in the cognitive domain, (2) severity of the deficits, (3) proportion of alterations to compensation process, and (4) inter-subject alterations of C-C chemokine receptor 5 (CCR5) expression after stroke [72,118]. Especially, poststroke CCR5 expression in the affected brain may be closely associated with the prognosis of motor and cognitive recovery after stroke. Pontine injury brings about secondary brain alterations remote from the damaged location during a term of diaschisis and is immediately followed by a compensation process for the diaschisis phenomenon. Very likely, after a stroke, diaschisis and compensation are mixed and, in some cases, competing with each other.

Another recent study investigated longitudinal regional cerebral blood flow (rCBF) changes in the acute phase, as well as follow-ups 1 week to 6 months after the pontine infarct (PI) [40]. There were significant rCBF decreases in the bilateral cerebellum and frontal (right supplementary motor area: SMA), parietal (right supramarginal gyrus), and occipital regions in the acute phase of the PI. The association of these alterations with the long-term cognitive outcome following a PI differed depending on the lesion location. In the left PI group, motor and memory recovery were associated with progressive increasing rCBF in the right supramarginal gyrus, whereas in the right PI group, memory and motor recovery were related to an increasing rCBF in the right SMA. 

### 4.5. Morphological and Neurodegenerative Changes Induced by Pontine Stroke: MRI Studies

The brain may change after a pontine ictus. Voxel-based morphometry (VBM) can detect morphological brain changes after a pontine infarction [41,43,119,120]. A previous study revealed that a pontine infarction may reduce the gray matter volume (GMV) in the cerebellum and expand the ipsilateral GMV in the middle frontal gyrus, middle temporal gyrus, mediodorsal thalamus, superior frontal gyrus, and contralateral precuneus [41]. It was suggested that GMV expansion in the ipsilateral mediodorsal thalamus was associated with motor recovery after a pontine infarction, although the association of GMV with cognitive recovery was not addressed. 

Diffusion tensor imaging (DTI) has been used to detect anterograde degeneration in the pyramidal tracts distal to a supratentorial lesion following an infarct [121]. Moreover, anterograde and retrograde degeneration remote from the primary lesion continuously deteriorates following a subcortical infarction, which interferes with poststroke functional recovery [45]. Likewise, DTI can detect the continuous deterioration of anterograde and retrograde degeneration in the pyramidal tract following a pontine infarct [45]. Therefore, it is plausible that similar progressive degeneration to that of the pyramidal tract following pontine injury may occur in any of the other tracts. In favor of this view, some functional connectivity MRI (fcMRI) studies have demonstrated alterations in the functional connectivity measures of the multiple pathways that are disrupted by focal damage to the pons [50,122,123]. The authors claimed that a pontine infarct may disrupt the prefrontal-cerebellar circuit. A decreased functional connectivity may be related to cognitive decline after a pontine stroke. 

### 4.6. Insights from Near-Infrared Spectroscopy

As mentioned above, our recent SPECT study demonstrated frontal hyper-perfusion and cerebellar hypo-perfusion, which shared consistent results from four patients with a pontine infarct [23]. In particular, the hyper-perfusion areas commonly included the supplementary motor area (SMA: Brodmann areas 6 and 8). Although the function of the SMA is not yet fully understood, the SMA may contribute to speech production [124,125,126], word retrieval difficulty by aging [127], inhibitory control [128], and executive function [129,130,131,132,133]. Previously, Penfield postulated that the SMA might be a third speech area, based on the evidence of vocalization and speech arrest by direct electric stimulations of the SMA [134,135]. The SMA is functionally divided into at least two distinct areas: the SMA proper, posterior to the vertical anterior commissorial (VAC) line, and perpendicular to the anterior commissure (AC)—posterior commissure (PC) plane; and the pre-SMA, anterior to the VAC line [136,137,138]. In fact, the SMA proper and pre-SMA are anatomically different: the SMA proper receives information from all components of the motor system [139], whereas the pre-SMA is densely interconnected with the prefrontal cortex (PFC) and also receives input from basal ganglia and cerebellum, but has no connection with the motor system [140].

To clarify the neurophysiology of SMA hyper-perfusion, we measured dynamic changes in the SMA responses during the phonemic verbal fluency task (VFT) as an index of executive function using functional near-infrared spectroscopy (f-NIRS) [23]. The pontine infarct group had no significant difference in their fluency ability compared with the age-matched control group. Furthermore, no significant differences in SMA responses could be detected between the two groups, but the SMA responses had a moderate correlation with the executive function. On the other hand, the pontine infarct group had executive dysfunction, as proven by the delayed TMT, expectedly making this domain relatively mild and transient. In fact, we observed the recovery of this domain a few months later. Intriguingly, a follow-up f-NIRS demonstrated increased the SMA responses coupled with improving the TMT-B, suggesting the contribution of the SMA to cognitive recovery after pontine injury [23]. 

### 4.7. Similarities and Differences of Functions among Pons, Cerebellum, and Thalamus 

D’aes and Mariën claimed that damage to the brainstem may affect cerebellar function, as cognitive decline due to a brainstem stroke seemed to share some common cognitive domains with the cerebellar cognitive affective syndrome (CCAS), comprising executive dysfunction, difficulties in spatial cognition, linguistic difficulties, and personality changes [15,19,141,142]. 

On the other hand, the function of the thalamus should be more complicated than those of the pons and cerebellum, and it would be different from those of pons or cerebellum in terms of cognition. The thalamus serves cognitive and language functions as the final hub of a sensory information relay to the neocortex, striatum, and hippocampus by divergent and convergent thalamocortical and corticothalamic pathways [143]. In a cognitive aspect, damage to the anterior portions of the thalamus generates memory loss [144,145], and damage to midline thalamic nuclei causes inattention and executive dysfunction [146,147]. In the linguistic aspect, it has been a matter of debate whether the thalamus plays a role in language [148,149]. According to a previous review [150], almost 90% of the left thalamic and bilateral thalamic patients 3 weeks to 4 months post-stroke presented with memory disturbances, inattention, executive dysfunctions, and behavioral and/or mood alterations. Linguistic difficulties, such as fluency (6.4%), repetition (15.1%), naming (72.2%), auditory comprehension (43.8%), reading (25%), and writing (65%) were found in patients with left thalamic lesions (n = 37), and comprehension (1/2), repetition (1/2), and naming (2/2) were found in those with bilateral thalamic lesions (n = 3). Our recent study reported that 25 of the 27 patients with acute thalamic stroke (92.6%) had cognition impairments, including inattention (18 patients), memory disturbances (15), executive dysfunctions (11), social behavioral disturbance (1), and aphasia (3) (Table 6). Also, we demonstrated that the thalamus plays a specific role in this loop, different from the pons or cerebellum, using two modalities of neuroimaging techniques such as SPECT and f-NIRS [151]. The SPECT results obtained from patients with thalamic lesions yielded evidence of common perfusion abnormalities in the fronto–parieto–cerebellar loop, including SMA, IFG, and surrounding language-relevant regions. In NIRS sessions during VFT, the thalamic stroke group encountered significant word retrieval difficulties relative to the age-matched healthy group. This implies that executive dysfunction due to a thalamic stroke may be more severe than a pontine ictus. Furthermore, a strong correlation between word retrieval and SMA responses has been demonstrated, and this suggests that there is a tight link between the thalamus and SMA. A follow-up NIRS revealed that increasing bilateral SMA responses may be associated with word retrieval improvements. The findings suggest that cognitive dysfunction after thalamic stroke may be related to the fronto–parieto–cerebellar loop, while language dysfunction is attributed to the SMA, inferior frontal gyrus (IFG), and language-related brain areas [151,152]. Together, the SMA may be responsible for the recovery of executive dysfunction after a thalamic stroke, as the SMA plays a role in cognitive recovery after a pontine stroke. These findings demonstrate that thalamic injury disrupts the SMA function more seriously than a pontine stroke, thus leading to cognitive impairments more directly than pons (Figure 3).

### 4.8. Limitations and Future Direction

Our observations in the case series are derived from only a small sample of patients at a single institute, resulting in the prevention of generalized results. Larger sample sizes recruited from multicenter institutes are needed to confirm these findings. In Japanese medical circumstances, where most of the patients are transferred out for intensive rehabilitation within a few weeks of admission, we have difficulty in obtaining data from the long-term follow-up neuropsychological evaluation. Therefore, the prognoses and long-term trajectories of the cognitive disabilities remains largely unknown. We need to establish a criterion to decide exactly which measure would be more sensitive and more specific to each domain (executive function, attention, and memory) among the in-depth neuropsychological tests. To date, executive function is composed of multiple cognitive components. Further study is required to clarify which component is more vulnerable for pontine injury. The function of pons has been paid no attention in the basic neuroscience field. In the coming decade, an advance in translational research using an animal model with invasive techniques would provide us with great details on the molecular mechanisms of cognition. For example, the acquired genetic manipulation of a specific molecule at a specific legion by means of “optogenetics” technology might help us better understand how pons would be involved in cognitive processing at the molecular level. The application of new techniques and translational research would provide us with a new pharmacological intervention or other therapeutic approaches and management strategies which might facilitate the improvement of cognitive disabilities. 

Pontine injury induces morphological and neurodegenerative changes in the fronto–ponto–cerebellar–thalamic loop, resulting in the failure of information processing and then leading to cognitive decline. f-NIRS could monitor the dynamic changes in the SMA associated with executive dysfunction due to pontine stroke and eventually, cognitive recovery by monitoring the SMA responses using f-NIRS and follow-ups. 

## 5. Conclusions

Nine personal observations and a review of the literature showed that a range of cognitive symptoms may result from isolated pontine damage. In particular, executive dysfunction represents the most common cognitive symptom. In the vast majority of the previous neuroimaging literature dealing with cognitive deficits after a pontine stroke, a frontal perfusion abnormality was found. In addition, our unique data combining SPECT and f-NIRS show the involvement of fronto–cerebellar diaschisis, and also suggest that SMA responses might eventually reflect the severity of cognitive decline due to pontine stroke and may also be related to the recovery. We finally posit that cognitive decline after a pontine stroke may be attributable to fronto–ponto–cerebellar diaschisis. In other words, pons constitutes an intrinsic part of the fronto–cerebellar–thalamic loop, while each area performs its part in cognitive processing in a hierarchical manner, and that an isolated pontine injury can result in a variety of symptoms that are typically associated with a disrupted processing relay.

## Figures and Tables

**Figure 1 biomedicines-12-00623-f001:**
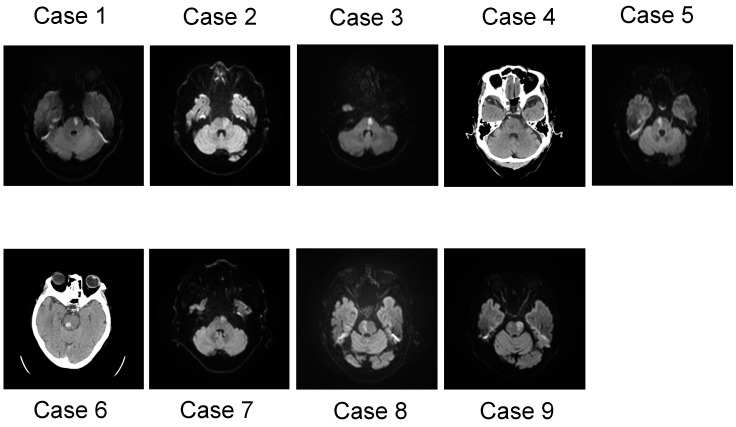
Distribution of strokes, which were detected using diffusion-weighted MRI and CT scans.

**Figure 2 biomedicines-12-00623-f002:**
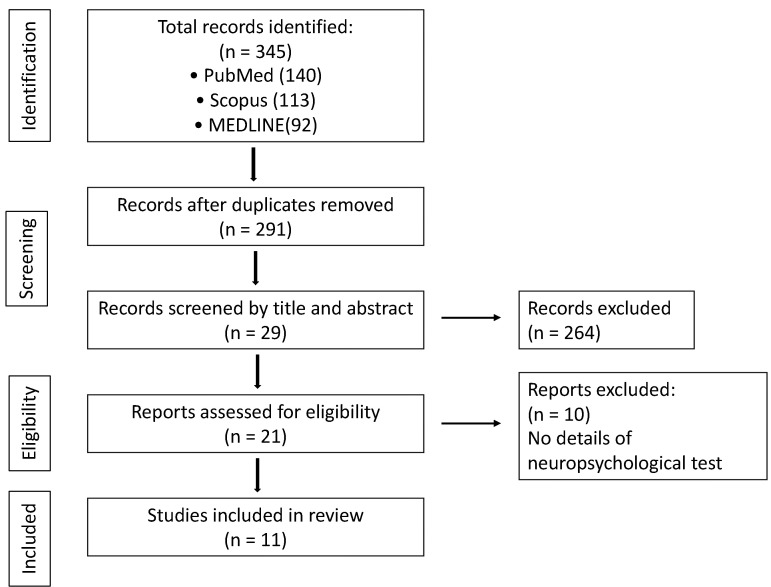
Flow diagram of included studies.

**Figure 3 biomedicines-12-00623-f003:**
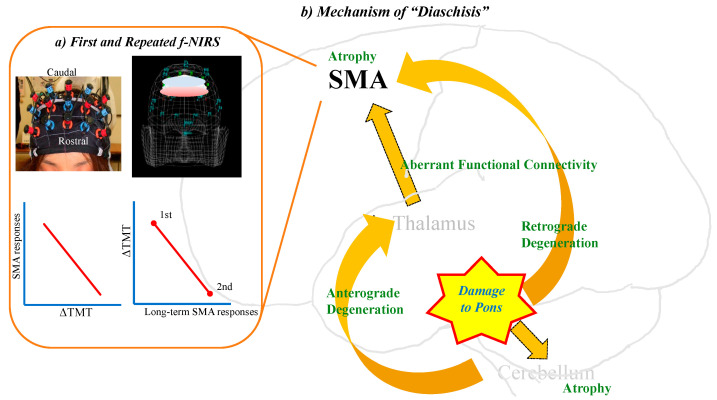
Neural mechanism underlying cognitive dysfunction after pontine stroke.

**Table 1 biomedicines-12-00623-t001:** Baseline characteristics of case series.

Case	Age	Gender	Type of Stroke	Laterality	Volume (mm^3^)	BRS on Admission	BRS at Discharge
Case 1	63	M	BAD	Left	273.5	6,5,6	6,6,6
Case 2	72	M	BAD	Left	1328.5	6,6,6	6,6,6
Case 3	75	F	BAD	Left	259	3,3,4	3,3,4
Case 4	76	M	Lacunar infarct	Median	325	No paresis	
Case 5	77	M	BAD	Right	873	4,4,5	6,6,6
Case 6	77	F	Hemorrhage	Right	870.5	6,6,6	6,6,6
Case 7	79	M	BAD	Left	372.5	No paresis	
Case 8	82	M	BAD	Right	1147.5	5,5,5	5,5,5
Case 9	86	F	BAD	Left	1126	2,2,4	3,4,3

M: male; F: female; BAD: branch atheromatous disease; BRS: Brunnstrom recovery stage.

**Table 2 biomedicines-12-00623-t002:** Cognitive domain profiles of our case series.

	Executive Dysfunction (FAB; TMT-B; ROCFT Copy)	Visual Memory Disturbance (ROCFT Recall)	Verbal Memory Disturbance (S-PA)	Inattention (TMT-A)
Case 1	+ (16/18; 106 s; 26/36)	+ (16/36)	+	− (43 s)
Case 2	+ 16/18; 156 s; 31/36	NA	NA	+ (110 s)
Case 3	+ (16/18; 116 s; 36/36)	+ (16/36)	+/−	− (42 s)
Case 4	+ (15/18; 292 s; 31/36)	+ (11/36)	+/−	+ (118 s)
Case 5	+ (14/18; 209 s; 20/36)	NA	−	− (60 s)
Case 6	+ (15/18; 79 s; 32/36)	+ (11/36)	NA	− (60 s)
Case 7	+ (14/18; 98 s; 34/36)	+ (16/36)	NA	+ (84 s)
Case 8	+ (10/18; NA; NA)	NA	NA	+ (243 s)
Case 9	+ (15/18; NA; NA)	NA	NA	NA

FAB: frontal assessment battery; TMT: trail making test; ROCFT: Ray–Osterrieth complex figure test; S-PA: standard verbal paired-associate learning test; +: present; −: absent; +/−: borderline; NA: not available.

**Table 3 biomedicines-12-00623-t003:** Risk of bias in the case-control studies.

Author	Year	Comparability of the Target Group	Target GroupSelection	Confounders	Measurement of Intervention/Exposure	Blinding of Assessors	Outcome Assessment	Incomplete Outcome Data	Selective Outcome Reporting
Van Zandvoort [60]	2003	Low	Low	Low	High	Low	Low	Low	Low
Fu [61]	2017	Low	Low	Low	High	Low	Low	Low	Low
Wang [48]	2022	Low	Low	Low	High	Low	Low	Low	Low

**Table 4 biomedicines-12-00623-t004:** Risk of bias in case reports and case series.

Author	Year	Documentation	Uniqueness	Educational Value	Objectivity	Interpretation
Our case series	Present	2	1	2	2	2
Hoffman and Watts [58]	1998	2	2	2	2	2
Hoffman and Malek [26]	2005	2	2	2	2	2
Garrard et al. [21]	2002	2	1	2	2	2
Nishio et al. [62]	2007	2	1	2	2	2
Maeshima et al. [63]	2010	2	1	2	2	2
D’aes and Marien [19]	2014	2	1	2	2	2
Neki et al. [64]	2014	2	1	2	2	2
Obayashi [23]	2019	2	1	2	2	2

Questions 1–5 comprise the tool for the risk of bias assessment for case reports and case series: 1. Did the patient(s) represent the whole case(s) of the medical center? (The studies did not mention whether the reported patient(s) represented the whole case(s) of the medical center, and we assumed that the authors have reported all the cases in their center given the rarity of this association.). 2. Was the diagnosis correctly made? 3. Were other important diagnoses excluded? 4. Were all important data cited in the report? 5. Was the outcome correctly ascertained?

**Table 5 biomedicines-12-00623-t005:** Summary of neuropsychological findings in the included studies.

Authors (Year) [Reference Number]	Classification	Executive Dysfunction	Inattention	Memory Disturbance	Linguistic Difficulty	Visuospatial Disability	General Intelligence
Hoffman and Watts (1998) [58]	5 cases	++	NA	±	NA	+	NA
Garrard et al. (2002) [21]	7 cases	++	+	+	−	−	±
Hoffman and Malek (2005) [26]	1 case	±	NA	−	+	+	NA
Nishio et al. (2007) [62]	1 case	++	+	±	−	−	±
Maeshima et al. (2010) [63]	1 case	+	+	+		−	−
D’aes and Marien (2014) [19]	3 cases	++	++	+	+	+	++
Neki et al. (2014) [64]	10 cases	+	NA	NA	NA	NA	+
Obayashi (2019) [23]	25 cases	+	++	+	−	NA	±
Van Zandvoort et al. (2003) [60]	Group comparison, 17 PS patients	+	+	NA	+	+	NA
Wang et al. (2022) [48]	Group comparison, 47 PS patients	++	−	+	NA	NA	NA
Fu et al. (2017) [61]	Group comparison, 34 PS patients	±	+	−	+	+	NA

PS: pontine stroke; ++: positive symptom; ±: suspected; −: negative.

**Table 6 biomedicines-12-00623-t006:** Similarities and differences: pons, cerebellum, and thalamus.

	Executive Dysfunction	Inattention	Memory Disturbance	Linguistic Difficulties	Spatial Cognition Difficulties	Personality Changes
Pons						
Cerebellum						
Thalamus						

The color density of the rows in each lesion represents the occurrence frequency of each domain, i.e., the deep green scale as a higher frequency and the light green as lower. The color gradient may represent hierarchical cognitive processing in the fronto–ponto–cerebellar–thalamic loop.

## Data Availability

The data are available upon reasonable email request.

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
