# Peer review of "Fronto–Cerebellar Diaschisis and Cognitive Dysfunction after Pontine Stroke: A Case Series and Systematic Review"

_biomedicines, 2024, doi:10.3390/biomedicines12030623_

Round 1

Reviewer 1 Report

Comments and Suggestions for Authors

The authors reported 9 pontine stroke cases and discussed available literatures about cognitive decline after pontine stroke and possible underlying mechanisms. The manuscript is well written. However, the manuscript can be improved if authors consider follow points.

1.    Please give the full name of f-NIRS when it is first appeared in the abstract.

2.    Please provide more information about long-term cognitive function of the 9 cases reported.

3.    Please provide a table to summarize the similarities and differences of pontine stroke and strokes occurred in other brain areas.

4.    It will be better if the authors can discuss more molecular mechanisms.

5.    Since this manuscript described 9 cases, information about informed consent statement should be provided.

Author Response

Comment 1.    Please give the full name of f-NIRS when it is first appeared in the abstract.

Response to comment 1.

We gave it (Page 2, line 84 ).

Comment 2.    Please provide more information about long-term cognitive function of the 9 cases reported. 

Response to Comment 2.

We have three cases with short-term follow-up investigation but no more information about long-term cognitive outcomes of the nine cases.

Comment 3.    Please provide a table to summarize the similarities and differences of pontine stroke and strokes occurred in other brain areas.

Response to comment 3.

Thank you for valuable suggestion. We provide a summary as table 2 (Page12 ).

Comment 4.    It will be better if the authors can discuss more molecular mechanisms.

Response to Comment 4.

We discussed molecular mechanism of cognition, especially executive function and memory. (page 8, lines 336-343; 359-361)

Comment 5.    Since this manuscript described 9 cases, information about informed consent statement should be provided.

Response to Comment 5.

Informed consent was obtained from all subjects involved in the study. (Page 13, line 575)

Reviewer 2 Report

Comments and Suggestions for Authors

The methodology in the submission is deficient. The authors include primarily elderly heterogeneous patients with what appears to be either a pontine infarct or pontine hematoma. Most patients test fairly well but there is no information about pre-stroke status and no identification of possible contributing factors to the cognitive deficit such as hippocampal atrophy, chronic white matter disease or ventricular enlargement which could explain the cognitive deficit reported. There is also the potential impact of post-stroke depression. The use of SPECT and f-NIRS are not clearly established as accurate indicators of the effect of acute stroke in my view. 

Author Response

Comment 1. The authors include primarily elderly heterogeneous patients with what appears to be either a pontine infarct or pontine hematoma.

Response to comment 1.

The heterogeneous patients with pontine stroke are commonly characterized by first-ever, isolated pontine injury.

Comment 2. Most patients test fairly well but there is no information about pre-stroke status and no identification of possible contributing factors to the cognitive deficit such as hippocampal atrophy, chronic white matter disease or ventricular enlargement which could explain the cognitive deficit reported.

Response to comment 2.

Thank you for valuable comments. It is true that our case series are selected referring to the inclusion criteria and the exclusion criteria. We described the criteria for detail (Page 2, lines 91- 96). We have promised no apparent morphological alterations by careful observation of their MRI and CT scans. 

Comment 3. There is also the potential impact of post-stroke depression.

Response to comment 3.

No participants are complained of any symptoms represented by post-stroke depression during hospitalization.

Comment 4. The use of SPECT and f-NIRS are not clearly established as accurate indicators of the effect of acute stroke in my view. 

Response to comment 4.

The use of these modalities is still spreading over the brain research field worldwide. The results of the database searched by “SPECT”, “brain”, and “cognition” as Keywords include 865 publications in PubMed. Also, the searching results from keywords of “Near infrared spectroscopy”, “brain”, and “cognition” showed 753 publications in PubMed. Especially, the relevant publications with NIRS are prominently growing. The fact implies that many researchers may rely on these techniques and become aware of the usefulness of f-NIRS and SPECT in clinical research field.

Reviewer 3 Report

Comments and Suggestions for Authors

The authors present the results of a single-center clinical study involving nine cases of isolated first-ever pontine stroke, in which neuropsychological findings were analyzed in depth and the relevant literature on cases describing cognitive disturbances following brainstem stroke lesions was identified. The authors found that pontine stroke could affect cognition, generally characterized by executive dysfunction. The authors suggest that cognitive decline after pontine injury represents a cerebro-cerebellar diaschisis. This report is potentially interesting, but the manuscript can be improved according to the following suggestions: 

1.In the Introduction it should be emphasized that recurrent lacunes are the most frequent cause of subcortical vascular cognitive impairment in acute lacunar stroke subtypes (Neuroepidemiology 2010;35:231-6). See and comment on this reference. 

2.Please include in the description of the nine patients of the study their clinical lacunar syndrome (pure motor hemiparesis, pure sensory syndrome, sensorimotor syndrome, ataxic-hemiparesis, dysarthria-clumsy hand, or atypical lacunar syndromes). 

3. The authors should clearly point out in the Discussion the relevance of clinically silent lacunes (as a major neuroimaging feature of cerebral small vessel disease) on cognitive performance. In a clinical study, more than half of the patients with a first-ever lacunar stroke had minor neuropsychological alterations. These minor alterations were mainly related to the presence of clinically silent lacunar infarcts on neuroimaging at this early stage of cerebral small vessel disease (see and add this reference BMC Neurol 2013; 13: 203). Did the authors consider this in their study protocol? 

Author Response

Our response to comment 1:

Concerning type of stroke, case series consisted of seven patients with pontine branch atheromatous disease (BAD), a patient with a paramedian lacunar infarct and a one with pontine hemorrhage. Our paper aims to explore the characteristics of cognitive decline after pontine damage and discuss the underlying neuropathology. On the other hand, above literature (Neuroepidemiology 2010;35:231-6) focused on risk factors, clinical features and outcome of first -ever cerebral lacunar infarcts (page 232, second paragraph), which seems not to be relevant to our current interest. Another paper as you mentioned below (BMC Neurol 2013; 13: 203) paid attention to the involvement of lacunar infarcts in cognitive impairment. This paper sought to investigate whether the pathology of lacunar infarcts could affect cognition of the patients, but patients with pontine lacunar was estimated as low OR (0.7) and the cognitive profiles were not discribed for detail.

Response to comment 2

A patient with paramedian lacunar infarct (case 4) presented dysarthria and right medial longitudinal fasciculus syndrome.

Our response to comment 3:

As shown in additional figure 1, case series consisted of seven patients with pontine branch atheromatous disease (BAD), a patient with a paramedian lacunar infarct and the remaining one with pontine hemorrhage.

Exclusion criteria was history of damage from stroke (cerebral infarct, cerebral hemorrhage, subarachnoid hemorrhage, and lacunar infarcts) (page 2, lines 93-94).

Reviewer 4 Report

Comments and Suggestions for Authors

This case report tried to interpret the associations between pontine stroke and cognitive decline as well as proposed a related hypothesis of cognitive deficits related to the fronto-potine-cerebellar-thalamic circuit in stroke along with the literature review.  It's an interesting topic. However, some concerns still need to be further addressed by the authors:

1. Since it is a case report, the author should provide more details of the 9 cases in the case description part: A) all the subjects underwent MRI, the authors should add the lesion images or mask overlay on the MRI images to show the exact lesion location. B) stroke side: left, right or bilateral and C) calculated lesion size for each subject (since the lesion size may related to the severity of the stroke and cognitive deficits) and stoke scale evaluation; D) The details information on the treatment.

2.  All 9 cases get a similar test battery, so it is better to organize the test score in a table. Or combine the information with Table 1. Are there follow-up data for the test?

3.  Most pontine stroke patients would get motor functions, did the author acquire the relation motor function test, such as Assessment of Sensorimotor Recovery After Stroke (FMA). 

Author Response

Response to comment 1.

Response to 1-A: We showed lesion images of all cases by adding figure 1.

Response to 1-B&C: We disclosed stroke side: left, right or bilateral and calculated lesion size for each subject by adding table 1.

Response to 1-D: The conservative treatment of each case for details was described.

Response to comment 2.

We showed test scores in Table 2. We could follow up test for only three cases and each was described in case series description.  

Response to comment 3.

We added table 1 to describe baseline characteristics of case series, including motor function as BRS.

Round 2

Reviewer 4 Report

Comments and Suggestions for Authors

The author already addresses most of my concerns, with only several minor comments for the current manuscript:

1. For Section 3.2, literature search,  please add the cutoff date for the literature search. 

2. Line 528, to be more precise, FC here should be resting state functional connectivity, and the term "long-term" should be avoided due to the ambiguous definition in context. 

Author Response

Response to Comment 1.

We added the cutoff date for the literature search.

Response to comment 2.

We think the part you pointed out may be line 485, not line528. The word “long term” was used in the paper cited, so we used it. But as you pointed out, “long term” is inappropriate. We deleted the word “long term”.